# Prevalence and Factors Related to Nomophobia: Arising Issues among Young Adults

Elissavet Vagka [1], Charalambos Gnardellis [2,*], Areti Lagiou [1] and Venetia Notara [1]

1 Department of Public and Community Health, School of Public Health, University of West Attica, 12243 Athens, Greece; elvagka@uniwa.gr (E.V.); alagiou@uniwa.gr (A.L.); vnotara@uniwa.gr (V.N.)
2 Department of Fisheries and Aquaculture, School of Agricultural Sciences, University of Patras, 30200 Messolonghi, Greece
* Correspondence: hgnardellis@upatras.gr

**Abstract:** Nomophobia is characterized as apprehension of being apart from smartphone, which causes the user to seek proximity with the device. The purpose of this study was to explore the prevalence and factors associated to nomophobia among young adults in Athens, the capital city of Greece. A cross-sectional study was performed on a sample of 1408 young adults aged 18–25 years. The questionnaire was anonymous, including the socio-demographic characteristics of the participants, the smartphone uses, and the nomophobia questions. Statistical analyses were done by simple univariable techniques or modeling the data through generalized linear models. Almost all participants (99.9%) exhibited any level of nomophobia, with the moderate level prevailing (57.0%). Women and non-working participants were more likely to exhibit severe nomophobia (adj PR = 1.57) and any level of nomophobia was 30% higher among the participants whose father had no academic degree ($p = 0.029$). In addition, 59% of those with severe nomophobia had very frequent phone checking ($p < 0.001$) while 45.8% with any level of nomophobia reported a negative influence on their academic performance. Attention should be paid to early prevention through the development of integrated health promotion programs.

**Keywords:** nomophobia; nomophobia questionnaire; young adults; smartphone; prevalence

## 1. Introduction

Mobile phones were introduced widely in the 1990s and since then they have been an integral element in daily living. From a luxury item, nowadays, they have been turned into an indispensable one. Smartphones, as part of the technological evolution, have opened a new era of research interest with respect to their impact on socio-emotional well-being [1].

In the early 21st century, a new term, nomophobia (NO Mobile PHOne phoBIA), was first introduced, as a finding of United Kingdom Post Office research, to describe the psychological effects of smartphone use [2]. In the forthcoming years, nomophobia has been regarded as the "disorder of modern world" and this term was used to describe the anxious feelings and their consequences in users due to the lack of smartphones and other communication equipment [3]. This situation was also described as the apprehension of being apart from smartphone, which causes the user to seek proximity with the device [4]. Therefore, psychological and health implications were positively linked with the increasing use of smartphones and nomophobia [5].

In the era of social media, individuals who have a fear of missing out on updates, social activities, and immediate rewards may display anxiety or problematic smartphone usage, known as nomophobia. It was found that college students primarily use social media apps on their smartphones [6]. High levels of social media usage were positively associated with nomophobia [7].

Common characteristics observed among nomophobic individuals include smartphone overuse, avoidance of restricted smartphone use areas, always carrying chargers,

owning a second smartphone, keeping phones nearby while sleeping, late-night smartphone use, and checking them immediately upon waking [8].

Those affected by nomophobia experience a fear of missing out on messages, events, and social media posts [9]. They feel anxious when they forget their smartphone or encounter low battery or network connection issues. Consequently, they tend to keep their phones switched on 24/7. The excessive obsession with smartphones among nomophobic individuals can significantly disrupt their daily behaviour [10]. Research conducted on university students indicated that excessive smartphone use, particularly for social networks, watching videos, and playing games, leads to significant time wastage and adversely affects academic performance. This results in decreased attention, bad grades, and increased university dropout rates [6].

In the last decade, the interest in exploring the new phenomenon of nomophobia has significantly increased. The researchers examined the prevalence and severity of nomophobia in association with several sociodemographic characteristics such as age, gender, duration and frequency of smartphone usage, academic performance, housing type, internet access, app usage, and parents' education level [11–13]. It is observed that nomophobia appears to be more prevalent in young adults [5]. It was stated that the age group most susceptible to nomophobia is those between 20 and 24 years old [14], as young individuals tend to adopt new technologies and tools more rapidly than others. Additionally, it was argued that smartphone use has become a dominant and defining characteristic for the younger generation with negative effects on psychological factors [15]. Smartphones have become an essential tool for constant socialization and communication among young people, but this dependency hinders their focus on crucial aspects such as personal development and health promotion [16].

Various studies indicated that females are more likely to exhibit nomophobia than males [17,18]. While no differences in problematic smartphone use between men and women was revealed [19], it was observed that women tend to demonstrate higher nomophobic levels which indicates a requirement for further research on gender differences [20].

In addition, individuals who spent many hours on their smartphones and owned them for several years showed elevated levels of nomophobia [21,22]. The rise of social networking as a dominant way of communication has raised concerns about excessive reliance on technology, especially among the younger generation [23]. Nomophobia, characterized by the fear of being without a mobile phone, has been proposed as a potential inclusion in the DSM-V, a diagnostic manual for psychiatric disorders [8]. It is crucial to conceptualize the effects of smartphone usage on individuals' overall well-being, given the significant influence that smartphones have in daily living and the possible consequences for frequent users if they are deprived of their devices [23].

Regarding parents' education level, a study claimed that the mean nomophobia score decreased according to the father's higher educational level [24]. Furthermore, a relevant study found that parents' educational level, the duration of smartphone use, and social feelings were significantly associated with the development of smartphone addiction [25]. Another study supported the negative associations between father's educational attainment and smartphone addiction, loneliness, and advancement motivation. The findings of the study demonstrated how risk variables, such as father's educational attainment, influenced individuals' excessive smartphone usage, which was strongly connected with smartphone addiction [26]. Additionally, another related study revealed a clear link between participants' smartphone addiction and their poor fathers' educational backround [27].

In Europe, there is a scarcity of findings on the prevalence and the effects of nomophobia [12,28,29]. To the best of our knowledge, there is no scientific evidence in Greece regarding the phenomenon of nomophobia among young adults and the related characteristics.

Therefore, the current study aimed to explore the prevalence and factors associated to nomophobia among young adults in Athens, the capital city of Greece.

## 2. Materials and Methods

### 2.1. Participants and Procedure

The participants were selected according to the inclusion requirements for the specific cross-sectional study, such as: (a) individuals' smartphone ownership, (b) being aged 18–25 years, and (c) completion of the informed consent form. Due to restricted access during COVID-19, the study sample was retrieved from 6 faculties of the University of West Attica and Post-Secondary Vocational Training schools located in Athens, the capital city of Greece.

For the analysis purposes, the participants were split into two age groups (i.e., 18–20 vs. 21+) so as to have a clear view of the differences between younger and older ages. Moreover, those aged up to 20 years are still in the phase of late adolescence; therefore, a difference exists as regards maturity and involvement with social media and smartphones. The two age groups were equally represented (i.e., 50.5% vs. 49.5%), which maximized the statistical power.

The study included 1408 male and female young adults aged 18 to 25 years, with a mean age of 20.7 years (SD = 2.0 years). The majority of the participants were women (71.7%) and university students (75.3%), while 31.5% were working. The questionnaire was anonymous and voluntary and was distributed during the lectures in the 2020–2021 academic year. Due to the pandemic restrictions, the study researcher supplied all necessary information and was accessible online throughout the questionnaire's completion via the Microsoft Teams platform; data were obtained electronically. The study was approved by the University of West Attica's research committee (14/21 September 2020) and was conducted in compliance with the Declaration of Helsinki (1989). Students were informed of the study's purpose and methods, and their consent was acquired.

### 2.2. Measures

The questionnaire consisted of three parts, including the: (a) socio-demographic characteristics such as age, gender, educational level, and parents' educational background; (b) smartphone use such as hours, calls, messages, and e-mails per day; and (c) nomophobia questionnaire.

Nomophobia Questionnaire (NMP-Q)

The 20-item Nomophobia Questionnaire (NMP-Q) has a 7-point Likert scale, with 1 representing "strongly disagree" and 7 representing "strongly agree". By summing the NMP-Q results, a numerical value between 20 and 140 was determined, with the highest score (NMP-Q = 140) indicating the most severe form of nomophobia. A score of 20 indicates nomophobia absence; a score of 21–59, mild nomophobia; scores of 60–99, moderate nomophobia; and scores of 100–140, severe nomophobia. In addition, the NMP-Q consists of four dimensions: (a) Not being able to communicate, (b) Losing connectedness, (c) Not being able to access information, and (d) Giving up convenience.

The original NMP-Questionnaire was developed by Yildirim and Correia (2015) [9] and validated for the Greek language. Exploratory and confirmatory factor analysis on the Greek questionnaire revealed a four-factor structure (subscales) in agreement with the original one. Moreover, a total nomophobia scale was assessed on the basis of all NMP-Q items [30]. The total scale presented a high internal consistency compared to the original NMP-Q (Cronbach alpha values are 0.945 for both for questionnaires). Moreover, the Cronbach alpha values for each factor were: (a) 0.936, (b) 0.895, (c) 0.867, and (d) 0.854, close to those of the original NMP-Q, which were 0.939, 0.827, 0.819, and 0.874, respectively.

### 2.3. Data Analysis and Statistical Methods

Statistical analyses were done by simple univariable techniques or modeling of the data through generalized linear models. Ordinal and nominal variables were presented as absolute and relative (%) frequencies. Association between nomophobia levels and sociodemographic characteristics of participants were evaluated through $\chi^2$ for a linear trend.

Continuous variables were given by their mean and median values, while comparisons between them, due to the skewed distributions and lack of equal variances assumption, were evaluated through a Kruskal–Wallis test.

Two generalized linear models were developed, having as response variables the total nomophobia scale in three and two categories. The first one was an ordinal logistic model with the total nomophobia score classified in three escalating categories (mild nomophobia = 21–59 of the total NMP-Q score, moderate nomophobia = 60–99 of NMP-Q, and severe nomophobia $\geq$ 100 of NMP-Q). The second model was a modified Poisson regression with, as a dependent binary variable, the mild/moderate nomophobia versus severe. A modified Poisson model was preferred instead of logistic regression to avoid inflated estimates of the prevalence ratios [31,32]. The predictor variables in the two models were the sociodemographic characteristics of the participants such as gender, age, education, working status, residency, nationality, and parents' educational level.

Results are presented as odds ratios (OR) and prevalence ratios (PR) along with their corresponding 95% confidence intervals (95% CI). In a first run, odds and prevalence ratios were estimated as unadjusted by univariable models and then adjusted for participants' sociodemographic characteristics. Statistical analyses were performed using SPSS v.28 statistical software (IBM Corp, Armonk, NY, USA).

## 3. Results

The prevalence of a mild level of nomophobia was 24.1% (339 pers), that of a moderate level was 57.0% (803 pers), and that of a severe level was 18.9% (266 pers). Only 2 participants out of 1408 showed low nomophobia. These individuals, in the analyses, were integrated into the mild category. Women and non-working participants were more likely to exhibit severe nomophobia (unadj PR = 1.63 and 1.42, respectively) compared to mild/moderate levels of nomophobia. An inverse association between age of participants, father's educational level, and severe nomophobia was observed. However, these two characteristics did not seem to be significant preconditions for severe nomophobia (Table 1).

To further evaluate the association between sociodemographic characteristics and nomophobia levels, adjusted odds ratios and prevalence ratios were estimated. Women and non-working participants had, respectively, 57% and 37% higher risk (adj PR = 1.57 and 1.37) to exhibit severe nomophobia compared to mild/moderate levels of nomophobia ($p$ values < 0.002 and 0.024 respectively). Even though the risk of exhibiting any level of nomophobia was 30% higher among the participants whose father had no academic degree ($p$ = 0.029), the risk of exhibiting severe vs. mild/moderate nomophobia was not significant ($p$ = 0.262) (Table 2).

Almost all participants (about 93%) who exhibited any level of nomophobia had a web connection in their smartphone ($p$ < 0.001). Of those with severe nomophobia, 59% had very frequent phone checking (up to 10 min) ($p$ < 0.001), while 45.8% of those with any level of nomophobia reported a negative impact on their academic performance. Participants with severe nomophobia had more expensive smartphones compared to those with mild and moderate cases ($p$ < 0.001). The main reasons reported for using smartphone were communication with family/friends (96.8%), news/information (90.8%), lessons (84.4%) and social media (81.3%). It should be noted that all the above percentages (with the exception of communication with family/friends) differed according to the level of nomophobia, i.e., there was a linear increase from mild to severe levels of nomophobia.

**Table 1.** Sociodemographic characteristics of study subjects by NMP categories.

| | | Nomophobia | | | | Nomophobia | |
| --- | --- | --- | --- | --- | --- | --- | --- |
| | | Mild N$_1$ (%) | Moderate N$_2$ (%) | Severe N$_3$ (%) | | Severe vs. Mild/Moderate | |
| N (%) | | 339 (24.1) | 803 (57.0) | 266 (18.9) | $\chi^2$ for linearity | unadj PR [1] | 95% CI PR |
| Gender | | | | | | | |
| Women | 1009 (71.7) | 216 (21.4) | 579 (57.4) | 214 (21.2) | <0.001 | 1.63 | 1.23–2.15 |
| Men | 399 (28.3) | 123 (30.8) | 224 (56.1) | 52 (13.0) | | | |
| Age groups | | | | | | | |
| 21+ | 697 (49.5) | 159 (22.4) | 405 (57.0) | 147 (20.7) | 0.043 | 1.21 | 0.98–1.51 |
| Education | | 180 (25.8) | 398 (57.1) | 119 (17.1) | | | |
| University | 1060 (75.3) | 230 (21.7) | 632 (59.6) | 198 (18.7) | 0.030 | 0.96 | 0.75–1.22 |
| Post-secondary | 348 (24.7) | 109 (31.3) | 171 (49.1) | 68 (19.5) | | | |
| Work | | | | | | | |
| No | 64 (68.5) | 211 (21.9) | 552 (57.3) | 201 (20.9) | <0.001 | 1.42 | 1.10–1.84 |
| Yes | 444 (31.5) | 128 (28.8) | 251 (56.5) | 65 (14.6) | | | |
| Residency | | | | | | | |
| With parents | 1045 (74.2) | 232 (22.2) | 623 (59.6) | 190 (18.2) | 0.256 | 0.87 | 0.69–1.10 |
| Alone | 63 (25.8) | 107 (29.5) | 180 (49.6) | 76 (20.9) | | | |
| Nationality | | | | | | | |
| Greek | 1319 (93.9) | 314 (23.8) | 757 (57.4) | 248 (18.8) | 0.779 | 0.89 | 0.58–1.36 |
| Other | 85 (6.1) | 24 (28.2) | 43 (50.6) | 18 (21.2) | | | |
| Father's Education | | | | | | | |
| Other | 934 (66.3) | 209 (22.4) | 537 (57.5) | 188 (20.1) | 0.018 | 1.22 | 0.96–1.56 |
| University | 474 (33.7) | 130 (27.4) | 266 (56.1) | 78 (16.5) | | | |
| Mother's Education | | | | | | | |
| Other | 804 (57.1) | 197 (24.5) | 447 (55.6) | 160 (19.9) | 0.700 | 1.13 | 0.91–1.42 |
| University | 604 (42.9) | 142 (23.5) | 356 (58.9) | 106 (17.5) | | | |

[1] Prevalence Ratio.

**Table 2.** Adjusted odds ratios and prevalence ratios derived from ordinal and Poisson regression analysis.

| | Ordinal Model | | | Poisson Model | | |
| --- | --- | --- | --- | --- | --- | --- |
| | Ordinal Scale of NMP | | | Severe vs. Low/Medium NMP | | |
| | adjOR | 95% CI OR | *p*-Value | adjPR [1] | 95% CI PR | *p*-Value |
| Gender | | | | | | |
| Women (vs. Men) | 1.65 | 1.32–2.08 | <0.001 | 1.57 | 1.19–2.08 | 0.002 |
| Age groups | | | | | | |
| 18–20 (vs. 21+) | 1.13 | 0.91–1.40 | 0.268 | 1.20 | 0.95–1.52 | 0.131 |
| Education | | | | | | |
| University (vs. post-secondary) | 1.25 | 0.97–1.59 | 0.080 | 0.93 | 0.72–1.19 | 0.547 |
| Work | | | | | | |
| No (vs. Yes) | 1.33 | 1.05–1.68 | 0.017 | 1.37 | 1.04–1.79 | 0.024 |
| Residency | | | | | | |
| With parents (vs. Alone) | 1.07 | 0.84–1.37 | 0.559 | 0.84 | 0.66–1.07 | 0.158 |
| Nationality | | | | | | |
| Greek (vs. other) | 1.04 | 0.68–1.61 | 0.852 | 0.91 | 0.60–1.40 | 0.678 |
| Father's Education | | | | | | |
| Other (vs. University) | 1.30 | 1.03–1.63 | 0.029 | 1.16 | 0.90–1.49 | 0.262 |
| Mother's Education | | | | | | |
| Other (vs. University) | 1.01 | 0.80–1.25 | 0.985 | 1.09 | 0.86–1.39 | 0.456 |

[1] Prevalence Ratio.

Additionally, the higher the level of nomophobia, the more likely the individual was to use a smartphone during daily activities (all *p* values ≤ 0.013). Quite interesting was that the highest presentence of the participants who used a smartphone while driving were those who demonstrated severe nomophobia (6%) (Table 3).

**Table 3.** Mobile phone use in percentages of study sample.

| | Nomophobia Categories | | | | $\chi^2$ for Linearity |
|---|---|---|---|---|---|
| | Mild N (%) | Moderate N (%) | Severe N (%) | Total N (%) | |
| Web connection in phone | 300 (88.5) | 751 (93.5) | 257 (96.6) | 1308 (92.9) | <0.001 |
| Checking | | | | | |
| Up to 10 min | 61 (18.0) | 290 (36.1) | 157 (59.0) | 508 (36.1) | <0.001 |
| 20 min | 47 (13.9) | 166 (20.7) | 51 (19.2) | 264 (18.8) | |
| 30 min | 67 (19.8) | 145 (18.1) | 25 (9.4) | 237 (16.8) | |
| >30 min | 164 (48.4) | 202 (25.2) | 33 (12.4) | 399 (28.3) | |
| Possession of second mobile phone | 52 (15.3) | 118 (14.7) | 43 (16.2) | 213 (15.1) | 0.816 |
| Cost of mobile phone | | | | | |
| <200 EUR | 199 (58.7) | 364 (45.3) | 85 (32.0) | 648 (46.0) | <0.001 |
| 200–400 EUR | 90 (26.5) | 260 (32.4) | 93 (35.0) | 443 (31.5) | |
| >400 EUR | 50 (14.7) | 179 (22.3) | 88 (33.0) | 317 (22.5) | |
| Affects academic performance | 130 (38.3) | 375 (46.7) | 140 (52.6) | 645 (45.8) | <0.001 |
| Reasons to use smartphone | | | | | |
| Communication with family/friends | 323 (95.3) | 784 (97.6) | 256 (96.2) | 1363 (96.8) | 0.395 |
| Mail | 249 (73.5) | 632 (78.7) | 224 (84.2) | 1105 (78.5) | 0.001 |
| Lessons | 277 (81.7) | 681 (84.8) | 230 (86.5) | 1188 (84.4) | 0.101 |
| Social Media | 241 (71.1) | 659 (82.1) | 244 (91.7) | 1144 (81.3) | <0.001 |
| Camera | 219 (64.6) | 587 (73.1) | 215 (80.8) | 1021 (72.5) | <0.001 |
| News/Information on the web | 296 (87.3) | 733 (91.3) | 250 (94.0) | 1279 (90.8) | 0.004 |
| When he/she uses smartphone | | | | | |
| Use/during eating | 93 (27.4) | 289 (36.2) | 139 (52.5) | 521 (37.2) | <0.001 |
| Use/during lessons | 106 (31.3) | 344 (42.8) | 144 (54.1) | 594 (42.2) | <0.001 |
| Use/during driving | 8 (2.4) | 15 (1.9) | 16 (6.0) | 39 (2.8) | 0.013 |
| Use/when he/she is with others | 106 (31.3) | 348 (43.3) | 160 (60.2) | 614 (43.6) | <0.001 |
| Use/in public transportation | 240 (70.8) | 669 (83.3) | 220 (82.7) | 1129 (80.2) | <0.001 |
| Use/when he/she is alone | 298 (87.9) | 764 (95.1) | 258 (97.0) | 1320 (93.8) | <0.001 |

Regarding the social networking, it was observed that participants with severe nomophobia, compared to those with mild and moderate, had more network friends and followers, they made more phone calls, and they spent more hours on the phone (all *p* values < 0.012), while, on the contrary, they spent less hours/week on the computer (although this difference is not significant) (Table 4).

**Table 4.** Mobile phone use by nomophobia categories.

| | Nomophobia Categories | | | | | | | | |
|---|---|---|---|---|---|---|---|---|---|
| | **Mild** | | **Moderate** | | **Severe** | | **Total** | | |
| | **Mean** | **Median** | **Mean** | **Median** | **Mean** | **Median** | **Mean** | **Median** | ***p* Value [1]** |
| Calls/day | 7.1 | 5 | 6.2 | 5 | 7.9 | 6 | 6.7 | 5 | 0.006 |
| Messages/day | 25.5 | 25 | 24.2 | 20 | 25.3 | 23 | 24.7 | 20 | 0.565 |
| Emails/day | 7.3 | 6 | 7.5 | 6 | 8.0 | 7 | 7.5 | 7 | 0.103 |
| Friends (Fb, MSN, games) | 1007 | 600 | 983 | 660 | 1125 | 856 | 1015.6 | 700 | 0.012 |
| Followers (Fb, Insta, Twitter) | 554 | 400 | 628 | 450 | 737 | 500 | 631.0 | 450 | 0.002 |
| Phone use hours/day | 5.8 | 5 | 6.7 | 6 | 7.9 | 7.5 | 6.7 | 6 | <0.001 |
| Computer use hours/week | 20.0 | 15 | 19.7 | 15 | 17.5 | 10 | 19.3 | 14 | 0.111 |

[1] Kruskal–Wallis non parametric test.

## 4. Discussion

Smartphones, as multifunctional devices, enable users to have access to a large number of applications. While it appears that the availability of smartphones benefits them, uncontrolled and excessive use may lead to negative outcomes [33].

In the present study, 1408 students participated. The majority were university students (75.3%), followed by post-secondary students (24.7%). It was found that almost all individuals demonstrated nomophobia, but the highest percentage was held by those who had a moderate level. Similarly, in a recent review study, nomophobia prevailed among young adults [34] and a number of studies reported a moderate level of nomophobia among university students [21,28,35–37]. However, an earlier study conducted by Yildirim et al. (2016) [18] found low prevalence among young adults (42.6%), which probably indicates that nomophobia is gradually expanding throughout the years.

Regarding gender, it was revealed that women had a higher level of nomophobia compared to men and greater odds to develop severe nomophobia. In terms of the scientific evidence, the results are controversial. Some studies are consistent with the findings of the present study [5,38,39], while others did not observe statistically significant difference between genders with regards to nomophobia levels [14,40–44]. Regardless of ambiguity, gender discrepancies could be explained by the fact that men and women seem to use their smartphones differently. For instance, men are more likely to use their smartphones for reasons related to work, whereas women primarily use them to communicate with loved ones [45].

The present study demonstrated an inverse relationship between age, father's education and levels of nomophobia; nevertheless, these two characteristics were not significant preconditions for severe nomophobia. In this line, it is also reported that individuals aged under 20 and 24 years had higher nomophobia levels compared to older ages [13,46,47]. On the contrary, other studies observed that age had no effects on participants' nomophobic behaviors [15,18]. Furthermore, a recent study pointed out that father's educational status was inversely linked to all nomophobia subscales [30]. Moreover, another study claimed the inverse association between nomophobia and father's educational level [38]. Nevertheless, since there is a lack of such evidence, further research is required to prove the association among father's educational level and nomophobia scores.

Almost all participants had a program for internet access via their smartphone (92.9%), and all of them exhibited some level of nomophobia (from mild to severe). Regarding the frequency with which the participants were checking their phone, 36.1% were checking up to every 10 min and 18.8% every 20 min. This conclusion corresponds to preceding research findings showing that university students check their smartphone more frequently [5,22,41,43]. Additionally, a high percentage of the participants were using their smartphone when they were alone (93.8%), as similarly observed in recent study (93.7%) [48]. Additionally, 45.8% of respondents believed that being preoccupied with their smartphone was an obstacle to their academic career. This percentage rises to 52.6% in those

with severe nomophobia. Accordingly, Qutishat et al. (2020) [11] revealed that students who experienced severe nomophobia reported poor academic achievement; however, this was not statistically significant.

The main reasons for using smartphones were communication with family/friends (96.8%), news/information (90.8%), lessons (84.4%), and social media (81.3%). Results from relevant studies demonstrated that the majority of the participants (92%) used their smartphones for social media, information (91.5%), calling, and sending SMS messages (87.6%) [49]. It is documented that individuals whose primary reason of use was social networking and texting had a higher risk of developing nomophobia [17,29,39].

Finally, the highest level of nomophobia was observed among participants with frequent smartphone use during their daily activities, which is also reported in a cross-sectional study which tried to explore the association between daily smartphone use and level of dependence [48]. An important finding, but not statistically significant, was that participants who spent more hours on the phone spent less hours/week on the computer. A possible implication is that young people nowadays are increasingly using mobile phones to access the Internet for most activities [50].

Smartphones are highly popular and represent a dominant piece of equipment among young adults. As a result, these devices heavily influence the way young people communicate. This phenomenon has significantly impacted the lives of many young adults, resulting in negative health outcomes and detrimental psychological effects [5].

Considering the increasing prevalence of information and communication technologies, further studies are needed to explore the phenomenon of nomophobia, particularly among younger generations, since limited research has been conducted in this area up to date. Health education and health promotion programs should be designed and implemented from early stages of life focused on the secure use of smartphones. It is also important that parents should participate in these programs so as to be informed about these issues. Young people should also take advantage of their free time by participating in sport activities or face-to-face interactions with their friends rather than using smartphones.

*Limitations*

It is rather difficult to generalize the results since the study was conducted among students from one university and Post-Secondary Vocational Training schools from Attica prefecture. However, the certain university is the third largest in Greece in terms of students' number and faculty. Another limitation is the unequal ratio of male and female respondents, which could greatly affect the results of the research, leading to gender bias. However, the results give an insight into the particular issue.

## 5. Conclusions

According to the study findings, almost all participants experienced some level of nomophobia. Nomophobia appears to be more prevalent in young adults, which lately is characterized as a "pandemic" problem among this age group. Individuals who are engaged in smartphone overuse are at a significant risk of developing nomophobic behaviours. Therefore, attention should be paid to early prevention through the development of integrated health promotion programs, even in primary school settings.

**Author Contributions:** Conceptualization, V.N., C.G. and E.V.; methodology, C.G., V.N. and E.V.; software, C.G.; formal analysis, C.G.; investigation, E.V.; data duration, E.V.; writing—original draft preparation, E.V.; writing—review and editing, E.V., C.G. and V.N.; supervision, C.G., V.N. and A.L. All authors have read and agreed to the published version of the manuscript.

**Funding:** This research received no external funding.

**Institutional Review Board Statement:** The study was authorized by the University of West Attica's research committee (14/21-09-2020) and was conducted in compliance with the Declaration of Helsinki (1989).

**Informed Consent Statement:** Informed consent was obtained from all subjects involved in the study.

**Data Availability Statement:** The data supporting the conclusions of this article will be made available by the authors, without undue reservation.

**Conflicts of Interest:** The authors declare no conflict of interest.

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
