# Peer review of "Prevalence and Factors Related to Nomophobia: Arising Issues among Young Adults"

_ejihpe, doi:10.3390/ejihpe13080107_

Round 1
Reviewer 1 Report
The authors performed a study to evaluate the prevalence and risk factors of nomophobia among young adults.
Idea: scientifically valid
English: needs minor revision
Title accepted
Abstract accepted
Introduction: needs more explanation of the knowledge gap. The authors did not discuss the problems associated with nomophobia that requires a thorough evaluation of its risk factors
Methods: Please elaborate more on how participants were selected.
Replace univariate by univariable.
Kruskal wallis test needs equal variance test. what test was performed to check for normal variance?
write the manufacturer of SPSS
Logistic regression is not suitable for more than two categories. It should be ordinal logistic regression.
Results:
Tabes should include the count in addition to the frequency.
How did you choose the variables for the multivariable regression?
Discussion
what are the future implications of this study?
What are the study limitations?
References: accepted
Needs minor revision
Reviewer 2 Report
Dear authors,
thank you for the opportunity to review your work.
Nomophobia is a completely new concept and thus your work contributes to a better understanding of this new concept.
But please provide explanations about some things:
In the part where you state the aim of the research, you state that you will deal with the prevalence of nomophobia among the population of Greek young people, but your research is limited to Athens, so it can give the wrong impression that you conducted the research on a sample of young people in the Greece.
It is not sufficiently clearly explained why you considered that the level of education of the parents as well as the level of education of the participants will be predictor variables for the explanation of nomophobia. Please provide explanation and give some theoretical insight.
It is not entirely clear why you divided the respondents into two groups of 18-20 years and 21+, please explain.
The NMP-Q questionnaire provides significantly more information such as Inability to access information, Denial of comfort, Inability to communicate, Loss of connectivity - why did you use only total score of the questionnaire? It will be good to see results on these variables in yours participants.
A major limitation of the research is the unequal ratio of male and female respondents, which could greatly affect the results of the research. Consider this limitation in the interpretation of your results.
In general, the part related to the limitations of the conducted research is missing.
Good luck.
Reviewer 3 Report
Dear Authors,
Thank you for the opportunity to read an interesting article entitled: Prevalence and factors related to nomophobia: arising issues among young adults'. The aim of this study was to analyse the prevalence and factors associated with nomophobia among students in Greece. The subject matter is interesting and extremely important looking at the growing problem of addiction to new technologies among young people around the world.
The strengths of the manuscript presented for evaluation are the very large sample size of over 1,000 respondents, the statistical analyses used and the citation of current literature mostly from the last 10 years.
The reviewer's job, on the other hand, is to help improve the article so that it meets the highest possible standards of the journal, therefore I will focus on its weaknesses.
- Subjects should be described in section 2.1 and not at the beginning of section 3.
- There are double spaces on many lines - e.g. line 13, 105, 147, 169, etc.
- In line 146, the designation of the 'p' factor should be in italics.
- In line 155, the full stop before the parenthesis is unnecessary.
Round 2
Reviewer 2 Report
Thank you for taking into consideration all suggestions.